# Topological edge states of interacting photon pairs emulated in a topolectrical circuit

Nikita A. Olekhno [1], Egor I. Kretov [1], Andrei A. Stepanenko [1], Polina A. Ivanova [1],
Vitaly V. Yaroshenko [1], Ekaterina M. Puhtina[1], Dmitry S. Filonov [2], Barbara Cappello [3],
Ladislau Matekovits [3] & Maxim A. Gorlach [1]✉

Topological physics opens up a plethora of exciting phenomena allowing to engineer disorder-robust unidirectional flows of light. Recent advances in topological protection of electromagnetic waves suggest that even richer functionalities can be achieved by realizing topological states of quantum light. This area, however, remains largely uncharted due to the number of experimental challenges. Here, we take an alternative route and design a classical structure based on topolectrical circuits which serves as a simulator of a quantum-optical one-dimensional system featuring the topological state of two photons induced by the effective photon-photon interaction. Employing the correspondence between the eigenstates of the original problem and circuit modes, we use the designed simulator to extract the frequencies of bulk and edge two-photon bound states and evaluate the topological invariant directly from the measurements. Furthermore, we perform a reconstruction of the two-photon probability distribution for the topological state associated with one of the circuit eigenmodes.

[1] Department of Physics and Engineering, ITMO University, Saint Petersburg 197101, Russia. [2] Center for Photonics and 2D Materials, Moscow Institute of Physics and Technology, Dolgoprudny 141700, Russia. [3] Department of Electronics and Telecommunications, Politecnico di Torino, I-10129 Torino, Italy. ✉email: m.gorlach@metalab.ifmo.ru

Recent years have brought a fast-paced development of topological physics in various systems ranging from traditional electronic setups[1–3] or cold atom ensembles[4–6] to mechanical[7,8], acoustic[9] and electromagnetic[10–14] structures governed by classical wave equations. A remarkable feature of such systems is the presence of unidirectional topological states robust against disorder and exhibiting zero reflection at sharp bends. In this context, photonic topological states appear to be especially attractive offering an energy-efficient alternative to their electronic counterparts, allowing for easier scaling, control, and manipulation and, ultimately, featuring a potential for on-chip integration[15–17].

While topological states of classical light are relatively well-studied, topological states of quantum light are much less explored with only few first studies currently available[16–20]. At the same time, topological states of quantum light can feature new exciting aspects of topological physics as, for instance, topological protection of biphoton correlations[17,20]. Further investigation of quantum-optical topological states may bring unexpected discoveries paving a way towards topologically protected quantum logic operations and quantum computations.

Nevertheless, the impact of interactions on quantum topological states remains largely uncharted, and first theoretical[21–25] and experimental[26–28] studies demonstrate rich manifestations of topology in interacting systems. The major challenge here is the difficulty in implementation of sizeable nonlinearities manifested already at the two-photon level[29].

As a particular example of few-body interacting system, we consider one-dimensional array of coupled nonlinear cavities described by the extended Bose–Hubbard model[30], illustrated in Fig. 1a. In this model, local photon–photon interactions give rise to exotic bound states of photon pairs persisting even in the case of repulsive nonlinearity[31,32] on which we focus from now on. Such repulsively bound pairs (doublons) were the subject of a series of recent theoretical[33–38] and experimental[39–43] studies. However, the observation of topological doublon edge states has remained elusive so far due to their absence in the standard Bose–Hubbard model[36]. Very recently, several realizations of doublon edge states in various modified models have been suggested[22,23,37,44,45], but their practical implementation still remained questionable.

Our present proposal relies on the extended version of Bose–Hubbard Hamiltonian

$$\hat{H} = \omega_0 \sum_m \hat{n}_m - J \sum_m (\hat{a}_m^\dagger \hat{a}_{m+1} + \hat{a}_{m+1}^\dagger \hat{a}_m) \\ + U \sum_m \hat{n}_m(\hat{n}_m - 1) + \frac{P}{2} \sum_m (\hat{a}_{2m-1}^\dagger \hat{a}_{2m-1}^\dagger \hat{a}_{2m} \hat{a}_{2m} + \text{H.c.}), \quad (1)$$

illustrated schematically in Fig. 1b. Here, the summation is performed over all resonators in the array enumerated with the index $m$, H.c. denotes the Hermitian conjugate of the term to the left, $\hat{a}_m^\dagger$ and $\hat{a}_m$ are the creation and annihilation operators for the photon at the $m$th resonator, and $\hat{n}_m = \hat{a}_m^\dagger \hat{a}_m$ is the photon number operator. The first term in the above Hamiltonian defines the energy of noninteracting photons in the resonator and has a trivial effect on the spectrum of photon pairs shifting it by $2\omega_0$. Therefore, for simplicity we use $2\omega_0$ as an energy reference for the

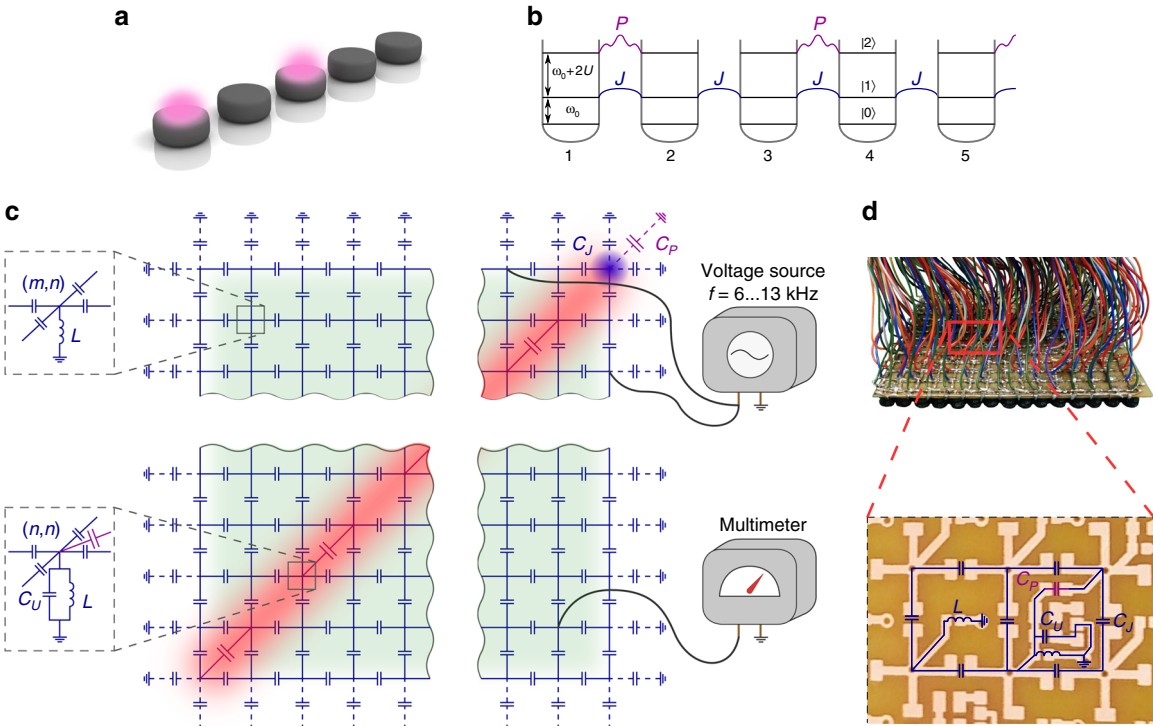

**Fig. 1 Two-photon one-dimensional quantum problem and its topolectrical circuit simulator. a** Artistic view of two-photon excitations in the array of microresonators with tunneling couplings. The depicted state is $\hat{a}_1^\dagger \hat{a}_3^\dagger |0\rangle$. **b** Extended version of Bose–Hubbard model considered in the present article. Single-photon tunnelings $J$ are shown by blue solid lines, direct two-photon tunnelings $P$ are indicated by purple wavy lines. **c** Top view of the equivalent two-dimensional topolectrical circuit with a voltage at the site $(m, n)$ corresponding to probability amplitude $\beta_{mn}$ for one photon to be located at the $m$th resonator of the array with another one located at the $n$th resonator [cf. Eq. (2)]. Colored regions show characteristic voltage patterns for two-photon scattering states (green), doublons (red), and doublon edge state (blue). External voltage source applied for the system excitation and voltmeter are shown to the right. Side view of the diagonal (lower inset) and off-diagonal (upper inset) sites of the topolectrical circuit, where grounding elements are shown. **d** The photograph of experimental setup having the size of 15 × 15 nodes. Inset shows the enlarged fragment of the circuit which includes two unit cells.

two-photon excitations and omit the corresponding term. The second term of the Hamiltonian describes single-photon tunneling between $m$th and $(m + 1)$st resonators in the array. Taken together, these two terms describe the linear array of identical cavities with the lowest eigenfrequency $\omega_0$ and the tunneling coupling $J$ between the nearest neighbors. The latter two terms of the Hamiltonian Eq. (1) describe effective photon–photon interactions mediated by the nonlinearity of the medium including an on-site photon–photon interaction $\propto U$ and a direct two-photon hopping $\propto P$, respectively.

A key property of this system is the emergence of bound two-photon topological states along with the trivial single-photon excitations. Hence, the topological order in this system is facilitated by the effective photon–photon interaction, which provides the simplest example of interaction-induced topological states of quantum light. Furthermore, in the strong interaction limit $U \gg J$ doublon excitations are described by the effective Su-Schrieffer-Heeger Hamiltonian[46], which is a paradigmatic one-dimensional topological model (see the analysis below and the Supplementary Note 1).

To overcome the difficulties with the direct experimental implementation of our model Eq. (1) and to bridge the gap between quantum-optical topological states and physics of interacting systems, we adopt the concept of topolectrical circuits[47–56] applying them to emulate an interacting two-body problem in one dimension. As detailed below, this approach is based on mathematically rigorous mapping of quantum two-body problem onto the classical setup of higher dimensionality, and this correspondence renders the topolectrical platform a powerful tool to study topological states of interacting photons.

## Results

**Topolectrical circuit realization and types of two-photon excitations**. Two-photon solutions to the stationary Schrödinger equation $\hat{H} |\psi\rangle = \varepsilon |\psi\rangle$ with the Hamiltonian Eq. (1) can be searched in the form

$$|\psi\rangle = \frac{1}{\sqrt{2}} \sum_{m,n=1}^{N} \beta_{mn} \hat{a}_m^\dagger \hat{a}_n^\dagger |0\rangle, \tag{2}$$

where $|0\rangle$ is the vacuum state and $N$ is the total number of resonators in the array. Superposition coefficients $\beta_{mn}$ characterize the probability amplitude for one photon to be present at site $m$ with the other one located at site $n$. Due to bosonic nature of the problem, superposition coefficients are symmetric, i.e. $\beta_{mn} = \beta_{nm}$.

The eigenvalue problem with the wave function Eq. (2) and the Hamiltonian Eq. (1) yields a linear system of equations with respect to the unknown coefficients $\beta_{mn}$, which can be written as

$$\sum_{m',n'} \left[ H_{mn,m'n'} - \varepsilon\, \delta_{mn,m'n'} \right] \beta_{m'n'} = 0. \tag{3}$$

Here, $H_{mn,m'n'}$ are matrix elements of the Hamiltonian in the basis Eq. (2), and $\varepsilon$ is energy variable.

To explore the physics of repulsively bound photon pairs and their edge states, we notice that the original one-dimensional two-particle problem is described by the same discrete wave equation as a two-dimensional tight-binding system. In such setup, vertical and horizontal links represent the hopping amplitudes for the first and second photons, additional links coupling the sites $(2n - 1, 2n - 1)$ and $(2n, 2n)$ emulate direct two-photon tunneling process, whereas eigenfrequency shift for the diagonal sites $(n, n)$ represents on-site interactions $U$ in the original quantum-optical problem (Fig. 1b).

The described tight-binding system can be readily implemented experimentally using arrays of coupled waveguides[57], coupled ring resonators[15,58], or even LC circuits[49]. Still, any such classical two-dimensional realization emulates a pair of distinguishable particles. In the latter case, the dynamics is governed by the first-quantized Hamiltonian

$$\begin{aligned} \hat{H} = &-J \sum_n \left( |x_n\rangle\langle x_{n+1}| + |y_n\rangle\langle y_{n+1}| + \text{H.c.} \right) \\ &+ 2U \sum_n |x_n, y_n\rangle\langle x_n, y_n| \\ &+ P \sum_n \left( |x_{2n-1}, y_{2n-1}\rangle\langle x_{2n}, y_{2n}| + \text{H.c.} \right), \end{aligned} \tag{4}$$

while the wave function in the first-quantized form reads:

$$|\psi\rangle = \sum_{m,n} \beta_{mn} |x_m, y_n\rangle, \tag{5}$$

where $1/\sqrt{2} \left[ \hat{a}_n^\dagger \right]^2 |0\rangle$ corresponds to $|x_n, y_n\rangle$, and $\hat{a}_m^\dagger \hat{a}_n^\dagger |0\rangle$ corresponds to the combination $1/\sqrt{2} \left( |x_m, y_n\rangle + |x_n, y_m\rangle \right)$ $(m \neq n)$. Note that the states with bosonic symmetry can be emulated exciting such a setup symmetrically with respect to the diagonal.

Choosing an appropriate platform, we aim not only to observe the excitation of doublon edge state, but also to reconstruct the associated probability distribution for the topological edge mode and to extract the topological invariant for bulk doublon bands directly from the experiment. Reaching this goal implies extensive measurements of field amplitudes at all relevant sites of the system for different excitation scenarios. Based on this, we choose the most accessible platform of topolectrical circuits, for which the node potentials $\varphi_{mn}$ correspond to the $\beta_{mn}$ coefficients in tight-binding equations.

To design the desired two-dimensional topolectrical circuit, we combine the first and the second Kirchoff's rules into the matrix equation

$$\sum_{m',n'} Y_{mn,m'n'} \, \varphi_{m'n'} = 0. \tag{6}$$

Here, every composite index $mn$ labels one site of the two-dimensional lattice having the coordinates $(m, n)$. Off-diagonal entries of the matrix $Y_{mn,m'n'}$ are equal to the admittances of the elements directly connecting the sites $(m, n)$ and $(m', n')$ in the circuit, while the diagonal elements are defined as

$$Y_{mn,mn} = -Y_{mn}^{(g)} - \sum_{(m',n')\neq(m,n)} Y_{mn,m'n'}, \tag{7}$$

where $Y_{mn}^{(g)}$ is the admittance of the element connecting site $(m, n)$ to the ground. Comparing Eqs. (3) and (6), we immediately recover that off-diagonal entries of the admittance matrix correspond to tunneling couplings $J$ or $P$ in the initial tight-binding model, while the diagonal elements are associated with the resonator detuning $U$. In addition, to realize the desired tight-binding model Eq. (3), the lack of neighbors for the edge or corner sites of a topolectrical circuit evident from Eq. (7) should be compensated by the proper adjustment of admittance $Y_{mn}^{(g)}$. Further analysis carried on in Supplementary Note 3 provides the following identification of tight-binding parameters in terms of circuit elements:

$$J = 1, \quad U = \frac{C_P + C_U}{2\,C_J}, \quad P = -\frac{C_P}{C_J}, \tag{8}$$

whereas the "energy" eigenvalue is inversely proportional to the

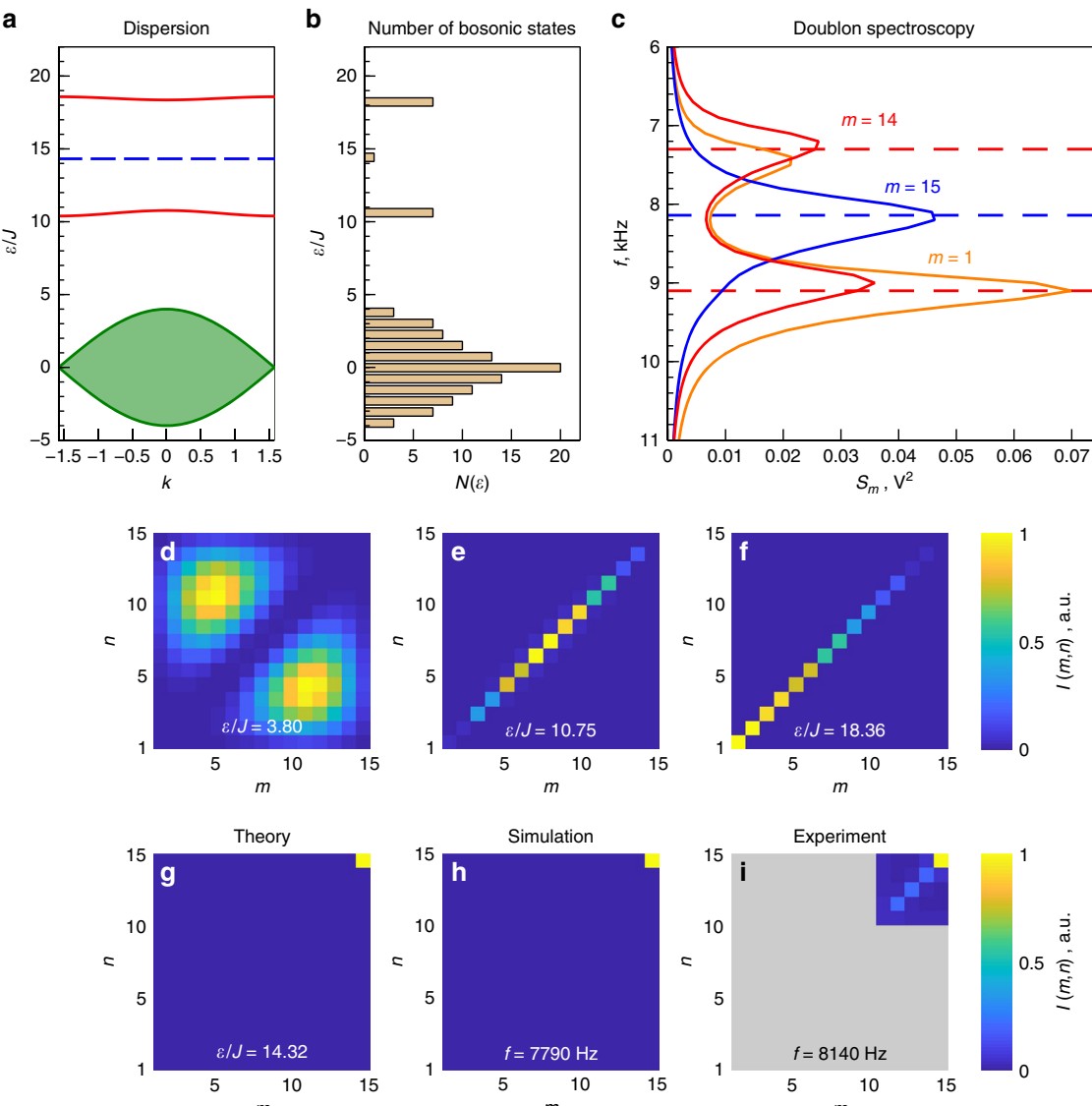

**Fig. 2 Theoretical studies and experimental emulation of two-photon excitations. a** Dispersion of two-photon eigenmodes calculated from the tight-binding model. Two red solid curves correspond to doublons, horizontal dashed line between them indicates the energy of the doublon edge state, and the shaded area at the bottom shows the continuum of two-photon scattering states. **b** Number of bosonic states in the tight-binding system of size 15 × 15 sites. **c** Doublon spectroscopy with quantity $S_m(f)$ [cf. Eq. (10)] determined from experimental data and plotted as a function of driving frequency. The feeding points (1, 1), (14, 14), and (15, 15) are labeled with corresponding values of $m = 1, 14, 15$ and are shown by orange, red, and blue solid lines, respectively. Characteristic peaks in the spectrum correspond to doublon modes. **d–g** Two-photon probability distributions $|\beta_{mn}|^2$ for the eigenmodes of 15 × 15 lattice described by the Hamiltonian Eq. (1) in the absence of disorder. Panels d, e, f and g correspond to the scattering states, lower and upper bulk doublon bands and doublon edge state, respectively. **h** Eigenmode reconstruction for a circuit of 15 × 15 nodes simulated taking into account the disorder in the element values as well as Ohmic losses (see "Methods" section for details). **i** Experimental implementation of the eigenmode reconstruction for the doublon edge state.

frequency $f$

$$\varepsilon = \frac{f_0^2}{f^2} - 4, \quad f_0^2 = \frac{1}{4\pi^2 LC_J}. \quad (9)$$

The scheme of the designed circuit is shown in Fig. 1c, while the photograph of the experimental sample is provided in Fig. 1d. For the experimental setup, the chosen element values result in $U = 7.09$, $P = -4.18$, which ensure that doublon bands are well separated from the continuum of scattering states and hence can be reliably detected.

Tight-binding calculations suggest that the eigenmodes supported by the designed structure can be classified into three types with the spectrum shown in Fig. 2a. The lower series of bands

shown by green is symmetric with respect to zero energy and corresponds to the two-photon scattering states. In the infinite-size limit, the photons in such states have energies in the continuous spectrum given by the sum of single-photon continuum energies. The typical probability distribution for the scattering state is depicted in Fig. 2d.

Two bands present at higher energies correspond to doublons. Probability distributions depicted in Fig. 2e, f suggest that the two photons most likely share the same resonator being free to move along the entire array. As a consequence, $\beta_{nn}$ coefficients are the dominant ones in the expansion of the two-photon wave function.

Finally, the gap between two bulk doublon modes is occupied by the doublon edge state with localization illustrated in Fig. 2g.

In the limit $U \gg J$ and $|P| \gg J$ the energy of doublon edge state scales as $2U$, whereas the splitting between the two bands is $\sim 2|P|$ (see Supplementary Note 1 for details). Thus, one can broadly tune the system behavior by varying the parameters $U$ and $P$ defined via the capacitances of circuit elements, Eq. (8).

In what follows, we focus on doublon states with a special emphasis on the doublon edge state which we prove to be topological. At first glance, such zero-dimensional localized state in a two-dimensional circuit may seem similar to higher-order topological states which have recently been predicted and observed in various systems[59–62] ranging from solid state[63] to photonics[64]. However, despite the seeming similarity, our proposal accesses completely different physics associated with quantum-optical topological states in interacting two-particle models emulated with the help of classical system of higher dimensionality. Quite remarkably, even in such situation topolectrical circuits provide a possibility to probe frequencies and probability distributions of bulk and edge doublon states, giving valuable information on the original 1D quantum problem.

**Experimental results**. To determine the spectral positions of doublon modes in the experiment, we apply voltage to one of the diagonal sites of the circuit, $(m, m)$, keeping the track of potentials $\varphi_{nn}^{mm}$ at all diagonal sites $(n, n)$. Next we construct the quantity[65]

$$S_m(f) = \sum_n |\varphi_{nn}^{mm}|^2, \tag{10}$$

which is evaluated as a function of driving frequency $f$. While scattering states feature zero overlap with the diagonal sites, doublon modes are characterized by the voltage maxima at the diagonal. Therefore, chosen excitation scenario selects just doublon modes, and their frequencies can be immediately associated with the characteristic peaks in the spectrum of $S_m$ clearly seen in Fig. 2c. Thus, $S_m(f)$ appears to be more convenient quantity compared with circuit impedance (Supplementary Note 7), since it filters out the contribution of the two-photon scattering states.

Experimental spectrum of $S_m$ plotted for $m = 1, 14$, and 15 in Fig. 2c features two characteristic peaks once the voltage is applied to the sites $(1, 1)$ or $(14, 14)$, whereas feeding of the site $(15, 15)$ results in a single peak. Based on our analytical model (see "Methods" section), we associate the former two peaks with bulk doublon modes, whereas the latter peak provides a clear signature of the doublon edge state. The respective eigenfrequencies are 7.28 and 9.08 kHz for bulk doublon modes and 8.14 kHz for the doublon edge state. A significant broadening of the peaks observed in Fig. 2c is caused by the combination of such factors as Ohmic losses inevitable in a real system and the dispersion of actual values of the circuit elements (see "Methods" for details).

The observed characteristic peaks in $S_m(f)$ provide an indirect evidence of bulk and edge doublon states. More information can be retrieved by measuring the voltage distribution at the nodes of the system at a given frequency. This distribution, however, depends strongly on the choice of the node $(m, m)$ at which the voltage is applied. In particular, exciting the system at $(15, 15)$ node, we recover the voltage pattern with a maximum at the point of excitation which can be explained either as doublon edge state or just as a trivial defect at the corner of the system.

To provide a clear evidence of doublon states independent of the choice of the feeding point, we have performed the reconstruction of voltage distribution for the doublon edge state eigenmode at the characteristic frequency of the corresponding peak. Such eigenmode tomography described in detail in Supplementary Note 2 employs the following steps. Throughout the whole procedure, the frequency of excitation, $f$, and the external voltage are fixed. For some symmetric choice of the

feeding points, e.g., $(m, n)$ and $(n, m)$, we measure full voltage distribution $\varphi_{m'n'}^{mn}$ at all nodes $(m', n')$ of the circuit and then evaluate the quantity

$$\Im(m, n) = \sum_{m', n'} |\varphi_{m'n'}^{mn}|^2. \tag{11}$$

Performing this step for various symmetric choices of the feeding points, we get an entire array of values for $\Im$, which is now considered as a discrete function of $m$ and $n$ coordinates. Finally, we depict this function on a colorplot Fig. 2i and observe a good agreement with the eigenmode distribution evaluated from the tight-binding model, Fig. 2g, as well as with the numerical solution of Kirchhoff's equations, Fig. 2h.

As shown in Supplementary Notes 2 and 4, the procedure outlined above reproduces the quantity

$$\Im(m, n) \propto \sum_k \frac{|\varphi_{mn}^{(k)}|^2}{(f - f_k)^2 + \gamma^2}, \tag{12}$$

where $\varphi_{mn}^{(k)}$ is the potential at site $(m, n)$ for the eigenmode with frequency $f_k$, and $\gamma$ is the effective dissipation rate related to Ohmic resistance of the circuit elements. Hence, $\Im(m, n)$ performs a sum over all eigenmodes with a Lorentzian-type weighting factor having a sharp maximum for the eigenmode with frequency $f_k$ matching the excitation frequency $f$.

Furthermore, Eq. (12) reveals that eigenmode tomography works exceptionally well when the spectral separation of the eigenmode (or a band of eigenmodes with a similar intensity pattern) from the rest of the modes exceeds the effective dissipation rate which is true for the doublon edge state, Fig. 2i. As a consequence, all key features of the corresponding state are well reproduced with only slight distortions present. On the other hand, all modes of the scattering continuum will feature much worse results of eigenmode tomography due to relatively large density of two-photon scattering states (Fig. 2b) and quite different field distributions for the different modes. At any case, such scattering states are not especially interesting since they feature neither topological protection, nor pronounced effects of interaction.

Experimental result of eigenmode tomography presented in Fig. 2i demonstrates good agreement with our theoretical expectations. Note also that while doublon edge state exhibits some hybridization with bulk doublons (Figs. 2e, f), it does not mix with the scattering states, which is a consequence of nonoverlapping spatial distributions of these modes and considerable spectral separation between them.

Having reconstructed the profile of the doublon edge eigenmode directly from the experimental data (Fig. 2i), we now turn to the discussion of its topological origin. Since the effective interaction strength $U$ is considerably larger than $J$, doublons are mostly localized at the diagonal of the 2D structure being effectively one-dimensional. Hence, the standard technique based on winding number calculation[66] can be applied.

Winding number in Su-Schrieffer-Heeger-type models is determined by plotting the ratio of voltages at the two sublattices (even and odd sites), $U_A(k)$ and $U_B(k)$, on the complex plane for a particular Bloch eigenmode, the topological scenario being characterized by the curve enclosing the coordinate origin. The winding number $W$ is defined as the number of revolutions of this curve around the coordinate origin. Note also that the winding number depends on the unit cell choice and therefore to reveal the topological edge state, the choice of the unit cell should be consistent with the array termination.

In the experiment, we set the excitation frequency to that of higher-frequency bulk doublon band, $f = 9550$ Hz, and measure the distribution of voltages at the diagonal keeping the track of

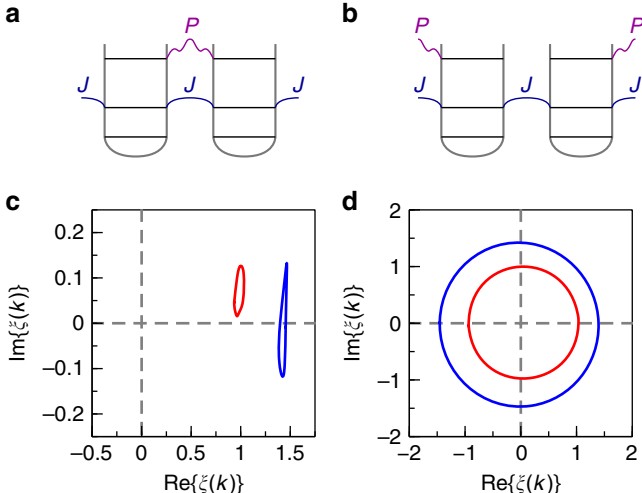

**Fig. 3 Topological characterization of doublon modes. a, b** Two possible choices of the unit cell. **c, d** The ratio of sublattice voltages $U_A(k)$ and $U_B(k)$, $\xi(k) = U_A(k)/U_B(k)$ plotted on a complex plane for the wave number $k$ varied over the entire Brillouin zone. Blue and red lines correspond to numerical simulation and experiment, respectively. The winding numbers $W = 0$ (**c**) and $W = 1$ (**d**) correspond to the unit cell choices shown in **a** and **b**. The voltage distribution is measured for the system excited at the node $(7, 7)$.

their phase. Recovering Bloch eigenmodes via Fourier transform and extracting sublattice voltages as outlined in "Methods" section, we finally get winding number graphs depicted in Fig. 3. Again, in a close agreement with our theoretical predictions, we recover that the unit cell choice without $P$ link inside gives rise to the nontrivial winding thus proving the topological origin of the observed doublon edge state. Note also that another choice of the unit cell with $P$ link inside yields zero winding number proving the absence of topological doublon edge state at the opposite $(1, 1)$ corner of the array.

## Discussion

To summarize, we have implemented a two-dimensional topolectrical circuit serving as an analog simulator for quantum one-dimensional two-particle interacting problem. Using the exact mapping of the initial quantum-optical system of two entangled interacting photons onto the classical setup, we have been able to simulate bulk and edge states of repulsively bound photon pairs.

Examining various excitation scenarios of the designed 2D circuit, we have not only reconstructed the doublon edge mode directly from experimental data, but also extracted the associated winding number from the measurements thus providing a rigorous proof of topological doublon edge state existence. Note that the possibility to conduct such extensive measurements is a distinctive feature of topolectrical platform, whereas the implementation of the same protocols for optical systems remains highly challenging.

The same approach can be used to emulate the dynamics of $N$ interacting particles in one spatial dimension. However, the dimensionality of the setup needed for that purpose should be equal to $N$ and therefore this emulation approach is meaningful only for reasonably low $N$ and only for the Hamiltonians conserving the number of particles.

Thus, we believe that our emulation of two-photon topological states induced by interactions uncovers new intriguing aspects of topological physics in interacting systems providing further insights into topological protection of quantum light.

## Methods

**Tight-binding equations.** The eigenmodes of quantum-optical problem under study correspond to the eigenvectors of the Hamiltonian Eq. (1). Since the Hamiltonian commutes with the operator $\sum_m \hat{n}_m$, the total number of photons is conserved. Hence, the wave function of arbitrary two-photon excitation can be represented in the form Eq. (2). Combining Eqs. (1) and (2) into an eigenvalue equation, we obtain the linear system of equations with respect to the unknown coefficients $\beta_{mn}$, which provide a probability amplitude for two photons to be present in $m$th and $n$th cavities:

$$(\varepsilon - 2U)\beta_{2m,2m} = -2J\left[\beta_{2m+1,2m} + \beta_{2m,2m-1}\right] + P\beta_{2m-1,2m-1}, \quad (13)$$

$$(\varepsilon - 2U)\beta_{2m-1,2m-1} = -2J\left[\beta_{2m,2m-1} + \beta_{2m-1,2m-2}\right] + P\beta_{2m,2m}, \quad (14)$$

$$\varepsilon\beta_{m,n} = -J\left[\beta_{m+1,n} + \beta_{m-1,n} + \beta_{m,n+1} + \beta_{m,n-1}\right], (m \neq n), \quad (15)$$

where $\varepsilon$ is the eigenmode energy, and $\beta_{mn} = \beta_{nm}$ due to the bosonic nature of the problem. The dispersion of two-photon modes and associated probability distributions plotted in the main text are calculated by solving the eigenvalue problem Eqs. (13)–(15).

**Excitation of the modes in the topolectrical circuit.** The eigenmodes of the circuit are found as the solutions to the eigenvalue problem Eq. (6). As further discussed in Supplementary Note 3, this eigenvalue problem has a one-to-one correspondence with the initial tight-binding system.

To describe the response of the circuit to the external excitation, we combine the first and the second Kirchhoff's rules to derive the equation with respect to the unknown potentials at the sites of the system. Excitation of the system is described as an external current injected symmetrically into $(p, q)$ and $(q, p)$ nodes of the circuit:

$$\sum_{m',n'} Y_{mn,m'n'}\,\varphi_{m'n'}^{pq} = \frac{I_0}{2}\left[\delta_{mp}\delta_{nq} + \delta_{mq}\delta_{np}\right]. \quad (16)$$

Inverting the matrix of admittances in Eq. (16), we immediately obtain the distribution of voltages at the nodes. To ensure reasonable results at resonances of the circuit, Ohmic losses should be properly taken into account. As further discussed in Supplementary Note 4, a topolectrical circuit with weak losses can be mapped onto the general driven-dissipative model studied in ref. [65].

Other important features of our topolectrical implementation include the negative sign of $P$ as guaranteed by Eq. (8). The designed system is mounted on FR4 substrate and includes the elements illustrated in Figs. 1c, d with the average values $L = 22.770\ \mu H$, $C_J = 1.002\ \mu F$, $C_U = 10.031\ \mu F$, and $C_P = 4.186\ \mu F$. Ohmic losses are introduced as a resistance attached in series to the inductance or capacitance. At frequencies of interest ($6.0 < f < 13.0$ kHz) such parasitic resistances are estimated as $R^L = 0.026$, $R_J^C = 0.2$, $R_U^C = 0.1$, and $R_P^C = 0.4$ Ohm based on specifications of the elements. In addition, we also incorporate the effects of disorder by reconstructing the entire map of the elements placed onto the fabricated circuit (Supplementary Note 5). The associated value distributions feature a broadening of up to 2%. Quite importantly, the parasitic resistance $R^L$ plays the major role at frequencies of interest. For that reason, designing our sample, we have chosen high-quality inductance coils with a sufficiently low dispersion in the values of inductance. There are other sources of disorder in the fabricated system as well, including nonideal contacts between soldered elements, parasitic inductances of tracks and capacitances created by the circuit board itself, and contacts between the board and the external measurement devices.

**Winding number.** To prove the topological nature of the doublon edge state, we apply the standard technique based on winding number evaluation[66]. To calculate the winding number for a chiral-symmetric system, one has to convert the Bloch Hamiltonian to the off-diagonal form

$$\hat{H}(k) = \begin{pmatrix} 0 & \hat{Q}(k) \\ \hat{Q}^\dagger(k) & 0 \end{pmatrix} \quad (17)$$

and then plot the curve for $\det \hat{Q}(k)$ on the complex plane with Bloch wave number $k$ spanning the entire Brillouin zone. The winding number is determined as a number of revolutions of the curve around the coordinate origin.

In the case of the Su-Schrieffer-Heeger model, Bloch Hamiltonian has the dimensions $2 \times 2$ taking the form Eq. (17) in the basis constructed from the modes of isolated resonators. Therefore, the scalar function $Q(k)$ is proportional to the ratio of voltages at the two sublattices (even sites and odd sites), which provides a relatively simple way to extract the winding number from experiment, as has been done e.g., in ref. [67].

In our case, chiral symmetry of doublon bands is only an approximation which holds in the limit $U \gg J$ only (see Supplementary Note 1 for details). In such a situation, one can neglect the mixing between the scattering continuum and doublon bands using an effective $2 \times 2$ doublon Bloch Hamiltonian. Furthermore, in such a case doublons are mostly localized at the diagonal, and therefore only the diagonal nodes of the circuit have to be examined.

Accordingly, we set the frequency of excitation to that of higher-frequency bulk doublon band, $f = 9550$ Hz, and measure voltage distribution at all diagonal nodes with an oscilloscope with a driving voltage applied to $(7, 7)$ node of the circuit. In the corresponding numerical simulation, we set excitation frequency to $f = 8730$ Hz to match the frequency of doublon peak as it appears in the simulations. The relative phase of voltages at the nodes is determined by measuring their time dependence and fitting the dependence by the sinusoidal function with the unknown phase. To get the results for a single Bloch mode, we perform a Fourier transform of the obtained voltages as follows:

$$U_A(k) = \sum_n U_A(n) \, e^{-ikn}, \quad U_B(k) = \sum_n U_B(n) \, e^{-ik(n-1)}, \tag{18}$$

where the indices $A$ and $B$ refer to the two different sublattices (even and odd sites). Next, we plot the ratio $\xi(k) = U_A(k)/U_B(k)$ on the complex plane for different choices of the unit cell with Bloch wave number spanning the range $(-\pi, \pi)$.

## Data availability

The data that support the findings of this study are available from the corresponding author upon request.

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

## Acknowledgements

We acknowledge valuable discussions with Alexander Poddubny, Alexander Khanikaev, Alexey Slobozhanyuk, and Evgenii Svechnikov. N.O. thanks Sergey Tarasenko for stimulating discussions on topological physics. Theoretical models were supported by the Russian Foundation for Basic Research (grant No. 18-29-20037), experimental studies were supported by the Russian Science Foundation (grant No. 16-19-10538). N.O. and M.G. acknowledge partial support by the Foundation for the Advancement of Theoretical Physics and Mathematics "Basis".

## Author contributions

M.G. conceived the idea and supervised the project. A.S., N.O. and M.G. worked out the theoretical models. N.O. and B.C. performed numerical simulations. N.O., M.G., and D.F. developed the circuit model and designed the experiment. E.K., D.F., V.Y., P.I., and E.P. fabricated the experimental setup. E.K., V.Y, N.O., P.I., and E.P. conducted the experiments. N.O., E.K., and B.C. supervised by L.M. performed post processing of the experimental data. N.O. and M.G. prepared the paper. All authors contributed extensively to the discussion of the results.

## Competing interests

The authors declare no competing interests.
