## [Peer Review File · Nature Communications]

Reviewers' comments:

Reviewer #1 (Remarks to the Author):

To the authors of " Topological edge states of interacting photon pairs realized in a topoelectrical circuit ":

In their manuscript, the authors simulate a system of two interacting photons whose doublons/repulsively bound states exhibit non-trivial topology, using electrical circuits. The experimental research presented seems solid, and is accompanied by a detailed theoretical analysis. The main results of the paper, in my opinion, may be summarised as (i) Experimental realisation of doublon edge states; (ii) experimental state tomography and (iii) experimental measurement of the winding number.

This is timely, important, novel and interesting research that deserves publication in Nature Communications. The ideas, execution and results of this work are of rather broad interest, including but not limited to other platforms such as cold atoms and quantum-optical systems. Moreover, there has been intense activity in few-body (in a particle language) effects in non-trivial topological systems recently and, as such, this work may trigger more interesting theoretical and experimental research on the subject, making it potentially very impactful.

The article is very well written, besides complete (including some useful additional information in the supplementary materials), and mostly gives fair credit to previous related work.

Before I can recommend the article for publication, I do have a few comments that should be addressed by the authors:

1 -- Critical comments

a) A surprising amount of literature in the classical-simulation community (mostly classical optics simulation of one and two-body physics) shows an unexplicable tendency to write second-quantised Hamiltonians when targetting two-body systems. This is the case here as well (Eq. 1). Although a priori this can be "corrected" with the right wording, it leads to confusing and potentially wrong statements. I have to say that the authors have been particularly careful about this, and therefore have made no mistakes. Also, the authors are aware of the fact that Hamiltonian (1) only applies to those eigenstates that are symmetric under exchange of the two "particles" coordinates (n,m) , but this is only noted in the supplementary materials. Writing the Hamiltonian in second quantisation without immediate explanation will only help spread confusion in the community. I would suggest (i) to write the Hamiltonian in first quantisation (e.g. [28]); (ii) specify clearly that the Hamiltonian is symmetric under exchange of (n,m) but that the system simulates **DISTINGUISHABLE** particles -- dynamically, two particles are effectively bosonic if and only if the initial state is bosonic; (iii) Figure 2b is either labelled as number of bosonic states, or the "fermionic" ones are added to the figure.

b) On Page 3, before the last paragraph, the discussion on scattering states is inaccurate. It may be rewritten to make it more rigorous. For example, from "The photons in such a state..." till the end of the paragraph could be replaced by:

"The photons in scattering states have energies in the continuous spectrum of the infinite-size limit, i.e. their energies correspond to the sum of single-photon continuum energies."

2 -- Minor comments

a) Reference [1] for condensed matter settings is a review, and so is Reference [3] for mechanical systems. However reference [2] is a research article (cold atoms). I would encourage the authors to cite more original work instead of reviews.

b) On Page 2, first sentence, references [26,27] are cited, while in the following sentence 28-33 are cited. I believe Ref. [28] is better suited as a reference together with [26,27], as it does not consider bulk and edge states of topologically non-trivial systems.

c) Together with references [14,15], cited on Page 1, I wrote a pre-print recently which is suitable for these platforms, arXiv:1907.08215, which may help motivate the article further.

d) I would suggest the authors to mention and cite a very remarkable paper on two atoms on a ladder by Greiner's group: Nature 546, 519 (2017). After all, this is a paper on few-body physics and some more background on the achievements in the field may be important.

e) On page 1, paragraph 1, the authors state that "photonic topological states" have the "potential for on-chip integration". Could the authors provide some explanation or a good reference?

Sincerely,
M. Valiente
Tsinghua University

Reviewer #2 (Remarks to the Author):

In their work "Topological edge states of interacting photon pairs realized in a topoelectrical circuit," Olekhno et al. experimentally study the two-particle bound-states of a 1D extended Bose-Hubbard model, with on-site and nearest neighbor interactions. They find two-particle bound-states that live at the edge and are spectroscopically isolated from other states.

The key to making this work go is the realization that a 1D model of two interacting particles on a lattice lives in a 2D Hilbert space, where one coordinate (say, X) is the location of one particle, and the other coordinate (say, Y) is the location of the other particle. If each particle can only hop to adjacent lattice sites, this corresponds to a discrete wave equation of a single particle on the 2D square lattice describing joint Hilbert space. The interesting point is that interactions between the particles, no-matter how strong, can be directly mapped into the 2D wave equation by shifting the relative energies of the lattice sites when $X=Y$, for onsite interactions, and $X=Y\pm 1$ for nearest-neighbor interactions. This means that rather than having to actually isolate individual photons (or other bosonic particles), make them interact with one another, and then detect their presence in the lattice, one can study the physics/dynamics of such a system using classical RF pulses propagating in a properly engineered 2D lattice.

This idea is fascinating, and, I think, teaches a typical reader quite a bit about the structure of Hilbert space and how strong interactions between particles work. I would very much like to see this work published in Nature Communications, as I think it is very interesting and insightful. The main improvement I would like to see is a bit more effort put into describing the mapping for the uninitiated reader: the technical details to unravel what is going on are all there, but the storytelling to explain it to the non-specialist is not. As this is so central to the interest of the work, more effort should be placed here. Along related lines, it is crucial to clarify the scaling with particle number: applying the method to more particles becomes exponentially harder: to study n particles requires an n -dimensional lattice – successfully explaining this to the reader will effectively convey what it is that makes quantum manybody physics so hard theoretically, and will provide a whiff of the flavor of what makes quantum computers so powerful!

There are a few key not-so-minor points that must (in my view) be addressed prior to publication:

- While this is absolutely an experimental study of the physics of pairs of interacting bosons (and I wonder if a pair of interacting fermions could be studied by modifying the connectivity of the

circuit? fascinating stuff!) what the authors observe is absolutely not bound states of photon pairs, as the title suggests: at least as best I can tell, the system has many many rf photons in it-- not two, even though it is simulating pairs of interacting bosons! To be fair, the authors have been very careful at the beginning of their manuscript to say that that they are studying the behavior of pairs of interacting photons, rather than the objects themselves. I expect this will be slightly confusing for a typical reader--the authors are actually doing (which is even more interesting, if you ask me!) is using non-interacting light to study the physics of interacting bosons. The argument that photons are bosons, so the authors can reasonably say that they are studying the physics of interacting photons, just adds confusion that obfuscates the beauty of the subtle mapping/Hilbert-space unwrapping that the authors harness to make their science go. Later, the authors state "Our observation of two-photon topological states induced by interactions..."--this, as best I can tell, is inaccurate: the authors simulate interactions by using a wholly linear system: they employ classical light pulses in a 2D inhomogeneous circuit to study pairs of interacting particles in a 1D homogeneous (modulo BCs) system. Let us not confuse what is simulated with what is actually observed. To my mind, this is all language and can be easily rectified, both in the body and in the title. A title like "Exploring topological states of interacting lattice bosons in a linear circuit" directly highlights the weirdness of the platform, leading the reader to immediately ask how this could be possible.

- The literature review is somewhat lacking:

- o I suggest referencing the review from Iacopo Carusotto and colleagues to cover most of the linear (and even some of the nonlinear) topological photonics: Ozawa, Tomoki, et al. "Topological photonics." *Reviews of Modern Physics* 91.1 (2019): 015006.

- o The statement "Nevertheless, the impact of interactions on topological states of quantum light remains almost fully uncharted with only theoretical studies available [14, 15]" misses some important work. Experimental studies exist and should be cited: the work of Roushan et al. explores interacting microwave photons on a single strongly interacting plaquette threaded with flux (Roushan, Pedram, et al. "Observation of topological transitions in interacting quantum circuits." *Nature* 515.7526 (2014)); the work of Clark et al. demonstrates two-photon Laughlin states of light created via Rydberg-mediated photon-photon scattering (Clark Logan W. et al. "Observation of Laughlin states made of light." arXiv:1907.05872 (2019)); the work of Barik et al. demonstrates a single non-linear emitter coupled to the edge of a topological lattice—a bit further removed but also relevant (<https://science.sciencemag.org/content/359/6376/666>)

- o It would also be good to cite the early theoretical and experimental work that initiated the field of topoelectric circuits, <https://doi.org/10.1103/PhysRevLett.114.173902> and <https://journals.aps.org/prx/abstract/10.1103/PhysRevX.5.021031>

- o I do not know if I missed anything else but it would be ideal if the authors could be a bit more thorough in their review of the literature.

Reviewer #3 (Remarks to the Author):

The authors study a very interesting topoelectrical circuit along the lines of Refs. [17,18...], but to judge on the validity of the model related to the experimental results, I have a few questions/suggestions for the authors.

In the Hamiltonian (1), the authors introduce the model including the new term "P", referred to as the double-hopping term. In the circuitry derivation, the authors mention $P = -C_p/C_J$, which then takes a negative value. Furthermore, the authors mention that in the experiments, $U=7.09$ and $P=-4.18$, related to Eqs. (6) and (7).

I have several questions & comments related to this step:

- In the Hamiltonian (1), the authors introduce the quantized version of the model in terms of harmonic oscillators. The capacitances C_j through a $(a+a^\dagger)_i * (a+a^\dagger)_j$ term could then also give a pairing term $a^\dagger_i a^\dagger_j$ term. In fact, this capacitance term is supposed to produce both the hopping term J and this new BCS like photon term. If the authors set $J=1$, can the authors justify why this BCS-term is not important? Or what is the value of w_0 in the Hamiltonian (1) compared to J [This is also related to Eqs. (S29) and (S30) ?

Along these lines, several works have shown the possible relevance of such a non-linear terms, for instance:

<https://journals.aps.org/prl/pdf/10.1103/PhysRevLett.118.197702>

<https://journals.aps.org/prb/abstract/10.1103/PhysRevB.93.020502>

In particular, the form of the $|GS\rangle$ may be different as the one in Eq. (2) of the authors and could allow a more general squeezed vacuum state:

<https://journals.aps.org/prb/abstract/10.1103/PhysRevB.97.041106>

in particular, if P is so large $P=-4.18$, can this form be more appropriate?

Related to this question, I tried to show why $P=-C_p/C_j$, but it is still not clear to me why P takes this form. Why is it judicious to take such large U values (through CU) and large values of $|P|$, compared to J .

Therefore: could the authors derive the quantized Hamiltonian (1) first, and justify the present form for the set of experimental parameters? The Kirchoff analogy in Eqs. (6) and (7) could also be introduced in a more pedagogical manner.

- Related to the measurement of S_m in Eq. (8), can the authors give more precision (related to Refs [47,17]) to justify why this measure is the appropriate one, for the doublon spectroscopy. The authors probe two-photon probability distribution, but with a particular choice of indices in the setting, in relation with Eqs. (8) and (10).

The authors write, page 3, " Probability distributions depicted in Figs. 2e, f suggest that the two photons most likely share the same resonator being free to move along the entire array. As a consequence, in the experimental structure, such modes are characterized by voltage maxima at the diagonal of the circuit." and then start a new discussion, which makes the introduction of Eq. (8) difficult to follow.

The doublon physics has also been studied in the Bose-Hubbard model (without the P term): <https://arxiv.org/pdf/0905.2963.pdf>, and the presence of edge modes can be detected through local density of states measurements.

- Related to the theory & simulation in Fig. 2, could the authors develop more the analysis related to Eq. (10). What is f_0 related to ϵ ? What is the definition of β in Fig. 5 of the Supplementary Material?

- In Fig. 3, can the authors explain the notations of what are the measured voltages U_A and U_B , here and what is the definition of the winding number W .

- On a general note, the authors mention in the introduction that this field of research is very active through Refs. [17-24], but does not emphasize much on why these new measurements would be sufficiently important to be published in Nature Communications.

Response letter

«Topological edge states of interacting photon pairs emulated in a topoelectrical circuit»

First of all, we would like to thank all Referees for thorough reading of our manuscript, their constructive comments and suggestions. We have revised our manuscript in accordance with the Referee's recommendations. For clarity, changed fragments of the text are highlighted by blue. Our point-by-point replies to the raised questions appear below.

Reviewer #1

Q1. A surprising amount of literature in the classical-simulation community (mostly classical optics simulation of one and two-body physics) shows an inexplicable tendency to write second-quantised Hamiltonians when targeting two-body systems. This is the case here as well (Eq. 1). Although a priori this can be "corrected" with the right wording, it leads to confusing and potentially wrong statements. I have to say that the authors have been particularly careful about this, and therefore have made no mistakes. Also, the authors are aware of the fact that Hamiltonian (1) only applies to those eigenstates that are symmetric under exchange of the two "particles" coordinates (n,m) , but this is only noted in the supplementary materials. Writing the Hamiltonian in second quantisation without immediate explanation will only help spread confusion in the community. I would suggest (i) to write the Hamiltonian in first quantisation, e.g. [Valiente, M. & Petrosyan, D. J. Phys. B 41, 161002 (2008)];

(ii) specify clearly that the Hamiltonian is symmetric under exchange of (n,m) but that the system simulates **DISTINGUISHABLE** particles -- dynamically, two particles are effectively bosonic if and only if the initial state is bosonic;

(iii) Figure 2b is either labeled as number of bosonic states, or the "fermionic" ones are added to the figure.

A1. We thank the Referee for this comment. The designed two-dimensional setup indeed emulates distinguishable particles, since in the general case voltages at the sites (m,n) and (n,m) can be different. Emulation of the states with bosonic symmetry in our setup is achieved only by applying the voltage symmetrically with respect to the diagonal which ensures excitation of the modes with symmetric profile.

To make this point clearer to the reader, we have revised the manuscript writing the Hamiltonian and the wave function in the first-quantized form, see p. 3, left column, Eqs. (4-5). There, we clearly specify that our setup emulates distinguishable particles in the general case. We have also revised Fig. 2b and its caption to highlight that we consider only the states with bosonic symmetry.

Q2. On Page 3, before the last paragraph, the discussion on scattering states is inaccurate. It may be rewritten to make it more rigorous. For example, from "The photons in such a state..." till the end of the paragraph could be replaced by: "The photons in scattering states have energies in the continuous spectrum of the infinite-size limit, i.e. their energies correspond to the sum of single-photon continuum energies."

A2. In order to clarify this point and make the discussion more rigorous, we have revised the manuscript text as suggested by the Referee: see the text highlighted by blue, p. 3, right column.

Q3. Reference [1] for condensed matter settings is a review, and so is Reference [3] for mechanical systems. However reference [2] is a research article (cold atoms). I would encourage the authors to cite more original work instead of reviews.

A3. We agree with the Referee's suggestion. In order to provide a fair credit to the previous works, we have cited original papers for all mentioned types of topological systems keeping a couple of references to review articles which might be useful for less expert readers.

Q4. On Page 2, first sentence, references [26,27] are cited, while in the following sentence 28-33 are cited. I believe Ref. [28] is better suited as a reference together with [26,27], as it does not consider bulk and edge states of topologically non-trivial systems.

A4. Following the Referee's recommendation, we have fixed the inaccuracy with citations.

Q5. Together with references [14,15], cited on Page 1, I wrote a pre-print recently which is suitable for these platforms, arXiv:1907.08215, which may help motivate the article further.

A5. We thank the Referee for providing the reference, we have cited it in the revised version of the manuscript as Ref. [25].

Q6. I would suggest the authors to mention and cite a very remarkable paper on two atoms on a ladder by Greiner's group: Nature 546, 519 (2017). After all, this is a paper on few-body physics and some more background on the achievements in the field may be important.

A6. We thank the Referee for suggesting this reference. To provide more background to our study, we have cited several relevant experimental papers on two-body physics in the introduction section (p. 1, left column). The paper by Greiner's group is added as Ref. [52].

Q7 On page 1, paragraph 1, the authors state that "photonic topological states" have the "potential for on-chip integration". Could the authors provide some explanation or a good reference?

A7. Presently, photonic topological states are successfully realized using the platform of silicon photonics which opens a route to fabricate photonic topological structures and meta-devices on a single chip. To clarify this, we have added experimental works Refs. [15-17] reporting the realization of topological structures on silicon photonics platform.

Reviewer #2

Q1. The main improvement I would like to see is a bit more effort put into describing the mapping for the uninitiated reader: the technical details to unravel what is going on are all there, but the storytelling to explain it to the non-specialist is not. As this is so central to the interest of the work, more effort should be placed here.

A1. We thank the Referee for thorough assessment of our work and positive evaluation. To make the paper more understandable to the broad audience, we have extended the part discussing the mapping of the original quantum-optical problem onto the classical setup. Newly added text is shown by blue in left column, p. 3.

Q2. Along related lines, it is crucial to clarify the scaling with particle number: applying the method to more particles becomes exponentially harder: to study n particles requires an n -dimensional lattice – successfully explaining this to the reader will effectively convey what it is that makes quantum many-body physics so hard theoretically, and will provide a whiff of the flavor of what makes quantum computers so powerful!

A2. We agree with the Referee that the discussion of N -particle case is quite relevant in the context of our work. Following this recommendation, we have discussed N -particle case and difficulties with its emulation in the discussion section, p. 5, right column.

Q3. While this is absolutely an experimental study of the physics of pairs of interacting bosons (and I wonder if a pair of interacting fermions could be studied by modifying the connectivity of the circuit? fascinating stuff!) what the authors observe is absolutely not bound states of photon pairs, as the title suggests: at least as best I can tell, the system has many many rf photons in it-- not two, even though it is simulating pairs of interacting bosons! To be fair, the authors have been very careful at the beginning of their manuscript to say that that they are studying the behavior of pairs of interacting photons, rather than the objects themselves. I expect this will be slightly confusing for a typical reader—the authors are actually doing (which is even more interesting, if you ask me!) is using non-interacting light to study the physics of interacting bosons. The argument that photons are bosons, so the authors can reasonably say that they are studying the physics of interacting photons, just adds confusion that obfuscates the beauty of the subtle mapping/Hilbert-space unwrapping that the authors harness to make their science go. Later, the authors state “Our observation of two-photon topological states induced by interactions...”—this, as best I can tell, is inaccurate: the authors simulate interactions by using a wholly linear system: they employ classical light pulses in a 2D inhomogeneous circuit to study pairs of interacting particles in a 1D homogeneous (modulo BCs) system. Let us not confuse what is simulated with what is actually observed. To my mind, this is all language and can be easily rectified, both in the body and in the title. A title like "Exploring topological states of interacting lattice bosons in a linear circuit" directly highlights the weirdness of the platform, leading the reader to immediately ask how this could be possible.

A3. We agree with the Referee that the title of the paper and some statements in the text may lead to the potential confusion since we indeed emulate two-photon topological states rather than observe them directly. In the revised version of the manuscript, we have corrected the title and these potentially confusing statements to avoid possible misunderstanding.

Q4. The literature review is somewhat lacking:

I suggest referencing the review from Iacopo Carusotto and colleagues to cover most of the linear (and even some of the nonlinear) topological photonics: Ozawa, Tomoki, et al. "Topological photonics." *Reviews of Modern Physics* 91.1 (2019): 015006.

The statement “Nevertheless, the impact of interactions on topological states of quantum light remains almost fully uncharted with only theoretical studies available [14, 15]” misses some important work. Experimental studies exist and should be cited: the work of Roushan et al. explores interacting microwave photons on a single strongly interacting plaquette threaded with flux (Roushan, Pedram, et al. "Observation of topological transitions in interacting quantum circuits." Nature 515.7526 (2014)); the work of Clark et al. demonstrates two-photon Laughlin states of light created via Rydberg-mediated photon-photon scattering (Clark Logan W. et al. "Observation of Laughlin states made of light." arXiv:1907.05872 (2019)); the work of Barik et al. demonstrates a single non-linear emitter coupled to the edge of a topological lattice—a bit further removed but also relevant (<https://science.sciencemag.org/content/359/6376/666>)

It would also be good to cite the early theoretical and experimental work that initiated the field of topoelectric circuits, <https://doi.org/10.1103/PhysRevLett.114.173902> and <https://journals.aps.org/prx/abstract/10.1103/PhysRevX.5.021031>

I do not know if I missed anything else but it would be ideal if the authors could be a bit more thorough in their review of the literature.

A4. We thank the Referee for pointing out these very relevant references. All suggested references are now incorporated into the revised manuscript in order to give a fair credit to the previous works. Additionally, we have cited several relevant theoretical and experimental papers including Refs. [25,27,52] of the revised manuscript.

Reviewer#3

Q1. The authors study a very interesting topoelectrical circuit along the lines of Refs. [17,18...], but to judge on the validity of the model related to the experimental results, I have a few questions/suggestions for the authors.

In the Hamiltonian (1), the authors introduce the model including the new term "P", referred to as the doublon-hopping term. In the circuitry derivation, the authors mention $P = -C_p/CJ$, which then takes a negative value. Furthermore, the authors mention that in the experiments, $U=7.09$ and $P=-4.18$, related to Eqs. (6) and (7). I have several questions & comments related to this step:

- In the Hamiltonian (1), the authors introduce the quantized version of the model in terms of harmonic oscillators. The capacitances CJ through a $(a+a^\dagger)_i * (a+a^\dagger)_j$ term could then also give a pairing term $a^\dagger_i a^\dagger_j$ term. In fact, this capacitance term is supposed to produce both the hopping term J and this new BCS like photon term. If the authors set $J=1$, can the authors justify why this BCS-term is not important? Or what is the value of w_0 in the Hamiltonian (1) compared to J [This is also related to Eqs. (S29) and (S30)?]

Along these lines, several works have shown the possible relevance of such a non-linear terms, for instance:

<https://journals.aps.org/prl/pdf/10.1103/PhysRevLett.118.197702>

<https://journals.aps.org/prb/abstract/10.1103/PhysRevB.93.020502>

In particular, the form of the $|GS\rangle$ may be different as the one in Eq. (2) of the authors and could allow a more general squeezed vacuum state:

<https://journals.aps.org/prb/abstract/10.1103/PhysRevB.97.041106>

in particular, if P is so large $P=-4.18$, can this form be more appropriate? Related to this question, I tried to show why $P=-C_p/CJ$, but it is still not clear to me why P takes this form. Why is it judicious to take such large U values (through CU) and large values of $|P|$, compared to J .

Therefore: could the authors derive the quantized Hamiltonian (1) first, and justify the present form for the set of experimental parameters? The Kirchoff analogy in Eqs. (6) and (7) could also be introduced in a more pedagogical manner.

A1. We thank the Referee for thorough reading of the manuscript and their comments. Below, we address the questions raised point by point.

First, we would like to highlight some conceptual differences between the physics of quantum LC circuits discussed e.g. in [Goren, *et al.* PRB 97, 041106(R) (2018)] and other papers mentioned by the Referee from one side and the physics studied in our work from the other. In contrast to the former case, we do not consider a quantized version of LC resonator array but rather study fully classical LC circuit. However, as we prove, this classical two-dimensional circuit precisely emulates the two-particle one-dimensional quantum-optical problem. The key idea here is one-to-one correspondence between Schrödinger equation for the two-photon wave function in Bose-Hubbard model [Eq. (1)] and Kirchhoff's equations describing the designed classical LC circuit. In this sense, we emulate topological interaction-induced states of photon pairs rather than observe them directly. In order to make this point clear to the reader, we have revised the title of the paper mentioning emulation of doublon states instead of realization. Second, we have described in greater detail the mapping of the original interacting quantum-optical problem onto the classical setup of higher dimensionality at p. 3, left column.

In this mapping, energy variable which enters Schrödinger equation for photon pairs is not equal to the frequency of the circuit eigenmode, but rather related to it in a more complicated manner as

given by Eq. (7). Furthermore, parameter ω_0 is not an important one and yields just a constant energy shift equal to $2\omega_0$ for all two-photon states. Therefore, without loss of generality, we count all energies from $2\omega_0$ energy level, as now indicated explicitly in the article text [p. 1, right column, after Eq. (1)].

To conclude, even though our experiments are fully classical, the reported results provide valuable insights into the physics of two-photon topological states in interacting systems, which has been appreciated by Reviewers #1 and #2.

Q2. Related to the measurement of S_m in Eq. (8), can the authors give more precision (related to Refs [47,17]) to justify why this measure is the appropriate one, for the doublon spectroscopy. The authors probe two-photon probability distribution, but with a particular choice of indices in the setting, in relation with Eqs. (8) and (10).

A2. We thank the Referee for raising this question. While the designed circuit supports quite large number of eigenmodes, we are mostly interested in a special kind of modes, namely, those which correspond to doublons (bound photon pairs) in the original quantum-optical problem. Such quasi-particles are characterized by the co-localization of the photons in the same cavity. Therefore, the respective modes of our 2D setup should exhibit voltage maxima at the diagonal.

In order to effectively couple to those modes, we excite our circuit at the diagonal. The resonances in the circuit response would correspond to the modes which have nonzero spatial overlap with the excitation point. Examining the calculated voltage distributions for the different types of modes, we recover that the modes which correspond to the scattering states have zero amplitude at the diagonal [see e.g. Fig. 2(d)]. Hence, our measurement protocol allows to selectively extract just the frequencies of doublon modes effectively excluding all other modes corresponding to the scattering states. For greater clarity, this is discussed in the revised text, p. 4, left column.

More detailed and quantitative discussion on the developed measurement protocols is provided in Sec. II and VII of Supplementary Materials.

Q3. The authors write, page 3, “Probability distributions depicted in Figs. 2e, f suggest that the two photons most likely share the same resonator being free to move along the entire array. As a consequence, in the experimental structure, such modes are characterized by voltage maxima at the diagonal of the circuit.” and then start a new discussion, which makes the introduction of Eq. (8) difficult to follow.

A3. We thank the Referee for spotting this issue. To make the text more coherent, we have reformulated this paragraph, p. 3, right column. In this part, we discuss the properties of two-photon eigenmodes of the original quantum-optical problem which actually coincide with symmetric eigenmodes of the designed circuit once energy variable ϵ is calculated in terms of eigenmode frequency f via the Eq. (9).

Q4. The doublon physics has also been studied in the Bose-Hubbard model (without the P term): <https://arxiv.org/pdf/0905.2963.pdf>, and the presence of edge modes can be detected through local density of states measurements.

A4. We thank the Referee for pointing out this reference. We have included it into the revised manuscript (Ref. [49]) in order to provide more background to this work.

Q5. Related to the theory & simulation in Fig. 2, could the authors develop more the analysis related to Eq. (10). What is f_0 related to ϵ ? What is the definition of β in Fig. 5 of the Supplementary Material?

A5. Equation (10) of the manuscript, basically, relates the quantity $J(m,n)$ measured in a real experiment to the intrinsic property of the system, namely, eigenmode profile $|\phi_{mn}(\mathbf{k})|$. Specifically, if the frequency of excitation matches the frequency of eigenmode, the measured pattern $J(m,n)$ will closely resemble the voltage distribution for the respective eigenmode.

Since the derivation of Eq. (10) is a bit lengthy and technical, for the readers convenience we have moved it to Sec. II of Supplementary Materials. Note also that the developed eigenmode tomography technique is quite general and applicable to a wide variety of Hermitian linear systems.

f_0 is a characteristic frequency of the designed circuit determined by capacitances and inductances of the elements as defined by Eq. (7). By choosing the appropriate circuit elements, we tune the resonance of our system to the desired frequency range which can be accessed with the available equipment.

β_{mn} are the superposition coefficients of the two-photon wave function defined in Eq. (2). Due to the employed mapping, these superposition coefficients correspond to the voltages in the nodes of the designed circuit for the respective eigenmodes.

In order to make these details clear to the reader, we have revised our manuscript accordingly and added the necessary explanations.

Q6. In Fig. 3, can the authors explain the notations of what are the measured voltages U_A and U_B , here and what is the definition of the winding number W .

A6. For clarity, we have explained these notations in the revised manuscript. Here, U_A and U_B are sublattice voltages, i.e. voltages measured at the diagonal (U_A refers to odd sites, U_B - to the even sites). We have also provided an explicit definition of the winding number W . For reader's convenience, further details on topological invariant evaluation can be found in the Methods section as well as Sec. X of Supplementary Materials.

Q7. On a general note, the authors mention in the introduction that this field of research is very active through Refs. [17-24], but does not emphasize much on why these new measurements would be sufficiently important to be published in Nature Communications.

A7. We believe that our work puts forward two major ideas which largely determine its novelty.

To the best of our knowledge, we propose the first example of a system which is topologically trivial in the single-photon regime hosting topological states of photon pairs. Thus, in contrast to the previous studies, our work provides the first evidence of interaction-induced topological states of quantum light promising novel avenues in quantum nanophotonics.

While the direct implementation of our proposal with qubit arrays or similar systems is highly nontrivial, we propose here an approach to emulate two-photon topological states using resonant LC circuits, which are easily accessible and allow to perform quite complicated measurement protocols.

Besides these ideas, we provide some further useful approaches on how to reconstruct the field distribution of the eigenmode directly from experimental data or retrieve the topological invariant from the measurements, which may be also appreciated by the community.

Finally, we would like to stress that while topological states of classical light are relatively well-established, topological states of quantum light remain largely uncharted. At the same time, this direction of research features a great potential for quantum information and quantum communications promising, for instance, topological protection of quantum entanglement. Therefore, as we believe, our work appears to be a timely contribution providing valuable insights into the emerging direction of quantum-optical topological states.

REVIEWERS' COMMENTS:

Reviewer #1 (Remarks to the Author):

The authors have considered all the issues I raised carefully and improved the readability and impact of the paper, I believe, significantly. I also think that they have improved the manuscript significantly thanks to the insightful comments of the other two reviewers. The article now looks and reads very "tight", and is as relevant and groundbreaking, if not more so, as the first version.

I strongly recommend this article for publication in Nature Communications.

Sincerely,
M. Valiente
Tsinghua University

Reviewer #2 (Remarks to the Author):

I am pleased to recommend publication of this manuscript in Nature Communications; the authors have satisfactorily addressed all of my concerns, particularly in emphasizing the distinction between the mapping and the physical platform!